# Flickering Emergences: The Question of Locality in Information-Theoretic Approaches to Emergence

**DOI:** 10.3390/e25010054

**Published:** 2022-12-28

**Authors:** Thomas F. Varley

**Affiliations:** 1Department of Psychological and Brain Sciences, Indiana University Bloomington, Bloomington, IN 47405, USA; tvarley@iu.edu; 2School of Informatics, Computing, & Engineering, Indiana University Bloomington, Bloomington, IN 47405, USA

**Keywords:** emergence, causality, higher-order interaction, partial information decomposition, synergy, networks, neuroscience

## Abstract

“Emergence”, the phenomenon where a complex system displays properties, behaviours, or dynamics not trivially reducible to its constituent elements, is one of the defining properties of complex systems. Recently, there has been a concerted effort to formally define emergence using the mathematical framework of information theory, which proposes that emergence can be understood in terms of how the states of wholes and parts collectively disclose information about the system’s collective future. In this paper, we show how a common, foundational component of information-theoretic approaches to emergence implies an inherent instability to emergent properties, which we call *flickering emergence*. A system may, on average, display a meaningful emergent property (be it an informative coarse-graining, or higher-order synergy), but for particular configurations, that emergent property falls apart and becomes misinformative. We show existence proofs that flickering emergence occurs in two different frameworks (one based on coarse-graining and another based on multivariate information decomposition) and argue that any approach based on temporal mutual information will display it. Finally, we argue that flickering emergence should not be a disqualifying property of any model of emergence, but that it should be accounted for when attempting to theorize about how emergence relates to practical models of the natural world.

## 1. Introduction

Despite the ubiquity of “emergence” in the natural world, a satisfying and broadly accepted formal definition of emergence has remained an outstanding problem in the field of complex systems. In the last decade, there has been a concerted push by multiple groups of theorists to develop a formal, mathematical approach to the question of emergence [1,2,3,4]. Many (although not all) of these approaches have used information theory [5] as the *lingua franca* on which to build their formal systems. We argue, however, that the specific applications of information to emergence bring with them curious consequences that have gone largely un-discussed and may have implications for how we think about emergent dynamics.

Essentially, every information-theoretic approach to emergence starts with the question of how the structure of a system in the present informs on the structure of the system in the future. There is an intuitive sense that, for very emergent systems, the “whole” somehow takes on a “life of its own” that is not trivially reducible to the dynamics of each element (in contrast to the way that the behaviour of an ideal gas can be trivially reduced to all of its elements, for example). There are a large number of ways to formalize this intuition: for example, Hoel et al. take a renormalization-based approach, comparing the full system to some coarse-grained micro-scale [6,7]. In contrast, Mediano, Rosas et al. consider joint-versus-marginal mereological relationships [8], and Chang et al. consider agent/environment distinctions and autopoetic notions of information closure [3]. These different approaches show how rich the space of putative “emergences” might be, even when starting from an apparently simple common starting point.

In addition to the common focus on how the present informs the future, another key, but rarely discussed commonality is that they are average measures, summarizing the expected value of a whole ensemble of possibilities in a single scalar value. This is a real limitation, as complex systems can hide a considerable amount of variability around the long-term mean (at the risk of being glib, the long-term, expected state of the Universe is one of bleak desolation and entropic void, but there’s quite a bit happening this week).

If, instead of considering the expected value over a whole distribution of configurations, one considers the local information contained in individual moments, they will see that the scalar, summary statistic can elide a large amount of variability. Sometimes, unexpected events can occur that break from the average behaviour of a system. For example, consider a flock of of starlings: on average, the dynamics of the flock show a strong, emergent structure, but when the birds unexpectedly scatter (perhaps in response to a predator), that emergence temporarily “breaks down” and must reform again in the future.

Written out in the language of information theory, these transient moments when the emergent dynamics break down can be understood as local instances of “misinformation” [9]—when knowing the present transiently stops helping you better predict the future (and sometimes can actually make your predictions worse). I call this *flickering emergence*: when the emergent structure “flickers” and the system briefly stops being a “whole” and becomes more like a collection of “parts”. As we will see, this possibility is an unavoidable consequence of any formal approach based on time-delayed mutual information. This phenomenon has not been discussed much in prior literature: one of the few instances was in the context of integrated information theory, where Hoel et al. showed that, for some systems, the integrated information could be greater at the macro-scale for some transitions, but not others [6]. The state-dependent nature of integrated information in this context could be seen as an early example of flickering emergence, although it relied on the specific (and somewhat baroque) mathematical machinery of integrated information theory and was not discussed as a general phenomenon. Similarly, Comolatti and Hoel [10] described a large number of state-dependent definitions of “causation” in the context of causal emergence, although the question of the stability of local emergence is not discussed.

## 2. Information Theory and Emergence

### 2.1. A Note on Notation

The information theoretic formalisms around emergence unfortunately require notation to represent a number of overlapping, and potentially confusing, concepts. In general, one-dimensional random variables will be represented with italicized uppercase letters (e.g., *X*). Specific realizations of that variable will be denoted with lower-case, italicized letters (e.g., X=x). The support set of *X* (i.e., the set of all states *X* can adopt) will be denoted with Fraktur font: X. We would read: $\sum_{x\in X}P\left(x\right)$ as “the sum of the probabilities of every state *x* that our random variable *X* can adopt”. Multidimensional variables will be denoted with boldface uppercase (e.g., **X**), their specific realizations as **x**, and their support sets in Fraktur font (X).

In addition to general and specific variables, there is also a need to differentiate between micro-scales and macro-scales. We will say that the *macro-scale* (after coarse-graining) of **X** is $\tilde{X}$, and the specific realizations are x and $\tilde{x}$ respectively. Only multivariate systems can be coarse-grained, and for our purposes, coarse-grained systems will always have at least two elements making them up. The expected value of a function will be denoted with calligraphic font (e.g., the expected mutual information between *X* and *Y* is I(X;Y)), while specific, local functions will be in italicized lowercase (e.g., the local mutual information between X=x and Y=y is i(x;y)). Finally, following [11], we will denote time with subscript indexing (i.e., $X_{t}$ is the random variable *X* at time *t*), and set membership with superscript indexing (i.e., $X^{k}$ is the $k^{\mathrm{th}}$ element of **X**). Both kinds of indexing may be used simultaneously.

### 2.2. Expected and Local Mutual Information

The core of almost all information-theoretic approaches to emergence has been to start with the *mutual information* between the past state of a system and its own future (sometimes called the *excess entropy* [12]). The mutual information is a fundamental measure of the (statistical) interaction between two variables [5]. For two variables *X* and *Y* with states drawn from support sets X and Y according to P(X) and P(Y):(1)$$\begin{array}{cc} I(X,Y) & =\sum_{\begin{array}{c} x\in X\\ y\in Y \end{array}}P\left(x,y\right)\log\left(\frac{P(x,y)}{P(x)P(y)}\right) \end{array}$$(2)$$\begin{array}{cc} \; & =H(X)-H(X|Y) \end{array}$$
where H is the Shannon entropy function. I(X;Y) quantifies how much knowing the state of *X* reduces our uncertainty about the state of *Y* on average (and vice versa, as it is a symmetric measure). Consequently, the mutual information is fundamentally about our ability as observers to make inferences about objects of study under conditions of uncertainty. Unlike more standard correlation measures, mutual information is non-parametric, sensitive to non-linear relationships, and strictly non-negative (i.e., knowing the state of *X* can never make us more uncertain about the state of *Y*). Being an average measure, I can also be understood as the *expected value* over a distribution of particular configurations:(3)$$I\left(X;Y\right)=E_{X,Y}\left[\log\left(\frac{P(x,y)}{P(x)P(y)}\right)\right]$$

We can “unroll” this expected value to get a *local* mutual information for every combination of realizations X=x,Y=y (sometimes referred to as the pointwise mutual information): (4)$$\begin{array}{cc} i(x;y) & =\log\left(\frac{P(x,y)}{P(x)P(y)}\right) \end{array}$$(5)$$\begin{array}{cc} \; & =h(x)-h(x|y) \end{array}$$

Unlike the average mutual information, the local mutual information can be negative: if P(x,y)<P(x)P(y) then i(x;y)<0. To build intuition, consider the case where I(X;Y)>0. On average, if we know that X=x, we will be better at correctly inferring the state of *Y* than if we were basing our prediction on *Y*’s statistics alone (and vice versa). Now, suppose we observe the particular configuration (x,y). What does it mean to say that i(x;y)<0 when I(X;Y)>0? It means that the particular configuration (x,y) that we are observing would be more likely if X⊥Y then if they are actually coupled (which we know, *a priori* that they are). Said differently, we would be more surprised to see X=x if we knew that Y=y than vice versa (see Equation (Equation 5)). *Local mutual information is negative when a system is transiently breaking from its own average, long-term behaviour.*

### 2.3. Temporal Mutual Information and Emergence

As mentioned above, a common starting point to assess formal theories of emergence is the time-delayed mutual information: information about the future that is disclosed by knowledge of the past. The total amount of information the entire past discloses about the entire future is the *excess entropy* [12]:(6)$$E\left(X\right)=I\left(X_{0:t-1};X_{t:\infty }\right)$$
where $X_{0:t-1}$ refers to the joint state of the entire past (from time t=0 to the immediate past) and $X_{t:\infty }$ refers to the entire future, from the present on. E(X) then provides a measure of the total dynamical “structure” of *X* (although it does not reveal how that structure is apportioned out over elements, see [11] for further discussion). Given the practical difficulties associated with infinitely long time series, it is common to assume that the system under study is Markovian and only “remembers” information from one time-step to the next. In this case, we would say that the constrained excess entropy is just:(7)$$E\left(X\right)=I\left(X_{t-1};X_{t}\right)$$

For a discrete system that can only adopt a finite number of states from the support set X, the temporal structure of the whole system can be represented in a *transition probability matrix* (Figure 1), which gives the conditional probability $P(x_{t}|x_{t-1})$ for every x∈X. Being a special case of the bivariate mutual information, the excess entropy can also be localized in the same way (and can also be either positive or negative):(8)$$e\left(x\right)=i\left(x_{t-1};x_{t}\right)$$

When e(x)<0, the particular transition $\left(x_{t-1};x_{t}\right)$ would be more likely to occur if subsequent moments were being drawn at random from some distribution P(X), rather than showing a temporal structure. If $P\left(x_{t}|x_{t-1}\right)<P\left(x_{t}\right)$, then you would be less likely to guess the correct $x_{t}$ if you knew the state $x_{t-1}$ then you would be if $x_{t}$ was being randomly selected from a 0-memory process. You would be more surprised to see $x_{t}|x_{t-1}$ than you otherwise would be. Your prediction of the future has been misinformed by the statistics of the evolving system. For formal theories of emergence that rely on the excess entropy, this kind of breakage from the system’s long-term expected statistics may represent a kind of failure mode whereby whatever “higher-order” dependency we are tracking is “interrupted”. The past transiently ceases to inform of the future and instead misinforms.

### 2.4. Two Formal Approaches to Emergence

Here we focus on two formal approaches to emergence: the coarse-graining approach first proposed by Hoel et al. [1], and an integrated information approach, from Mediano, Rosas et al. [13]. We chose these two since, despite using much of the same mathematical machinery to answer a common question, they lead to very different interpretations of what emergence is. We should briefly note, however, that these are not the only information-theoretic formal approaches to emergence, for example Barnett and Seth have a proposal based on dynamical independence between micro- and macro-scales [4], and Chang et al. proposed a theory based on scale-specific information closure [3,14]. Both of these frameworks are based on excess entropy, or slight modifications thereof (Barnett and Seth consider the temporal conditional mutual information, for example). Based on the use of temporal mutual information, we anticipate that incongruous dynamics and flickering emergence are likely to appear in local formulations of both approaches (barring an unexpected mathematical result).

#### 2.4.1. Coarse-Graining Approaches to Emergence

This framework was one of the first explicit, formal information-theoretic approaches to emergence, and remains one of the most well-developed. The coarse-graining approach compares the informational properties of a system **X** with the properties of a dimensionally reduced model $\tilde{X}$. The core measure is the effective information, which is the excess entropy with a maximum-entropy distribution forced on the distribution of past states:(9)$$\begin{array}{cc} F(X) & =E{\left(X\right)}_{|H\left(X_{t-1}\right)=H^{\max}} \end{array}$$(10)$$\begin{array}{cc} \; & =I(X_{t-1}^{H_{\max}};X_{t}) \end{array}$$

The effective information quantifies how much knowing the past reduces your uncertainty about the future if all past states are equally likely. Consequently, it is a statistical approximation of experimental intervention: by forcing the prior distribution to be flat, F(X) is not confounded by the potential for biases introduced by an inhomogeneous distribution $P(X_{t-1})$. F is bounded from above by log(N) (where N=|X|): if F(X)=log(N), then knowing $X_{t-1}$ completely resolves all uncertainty about the future (every $x_{t-1}$ deterministically leads to a unique $x_{t}$). In contrast, if F(X)=0, then knowing the past reduces no uncertainty about the future. This bound allows to normalize F to the interval [0,1], which we refer to as the *effectiveness* ($\bar{F}\left(X\right)$):(11)$$\bar{F}\left(X\right)=\frac{F(X)}{\log(N)}$$

Hoel et al. [1] claim that “emergence” occurs when, for system **X**, there exists some coarse-graining $\tilde{X}$ such that:(12)$$\log\left(\frac{\bar{F}\left(\tilde{X}\right)}{\bar{F}\left(X\right)}\right)>0$$

In this case, the macro-scale is more effective than the micro-scale: knowing the past of the macro-scale resolves a greater proportion of the uncertainty about the future of the macro-scale than it would when using the “full”, micro-scale model.

Consider the example system displayed in Figure 1: here, a 4-element, Boolean network evolves according to the micro-scale transition probability matrix. The system is then bipartitioned and each pair of elements (indicated by colour) is aggregated into a macro-scale with a lossy logical function (logical-AND), resulting in the 2-element system. Crucially, X is a system that displays non-trivial emergence upon coarse-graining into $\tilde{X}$: $\log(\bar{F}\left(\tilde{X}\right)/\bar{F}\left(X\right))\approx 0.202$.

#### 2.4.2. Synergy-Based Approaches to Emergence

The synergy-based approach to emergence takes a different tack when looking for emergent phenomena in multi-element, dynamical systems. Where the coarse-graining emergence considers the relationship between scales, the integrated emergence looks at the relationships between “wholes” and “parts” at a single scale. Once again, the primary measure is the excess entropy (a maximum entropy prior is not explicitly assumed, although it is an option [13]). We begin with the insight that, for a multivariate system **X**, E(X) gives a measure of the total temporal information structure of the whole, but it says nothing about how that information flow is apportioned out over various elements of X. For example, if every element of X is independent of every other the excess entropy can still be greater than zero if the excess entropy of each part is greater than zero:(13)$$\;E\left(X\right)=\sum_{i=1}^{\left|X\right|}E\left(X^{i}\right)⟺X^{i}⊥X^{j}\forall X^{i},X^{j}\in X$$

In this case of a totally disintegrated system, then $E\left(X\right)-\sum_{i=1}^{\left|X\right|}E\left(X^{i}\right)=0$: the problem of predicting the future of the “whole” trivially reduces to predicting the futures of each of the “parts” considered individually. This insight prompted Balduzzi and Tononi to propose a heuristic measure of “integration” (the degree to which the whole is greater than the sum of its parts), which we refer to as Φ(X):(14)$$\;\Phi \left(X\right)=E\left(X\right)-\sum_{i=1}^{\left|X\right|}E\left(X_{i}\right)$$

If Φ(X)>0, then there is some predictive information in the joint state of the whole **X** that is not accessible when considering all of the parts individually. This is a rough definition of “integrated emergence”: when the future of the whole can only be predicted by considering the whole qua itself and not any of its constituent parts.

Φ(X) is only a rough heuristic though, and it can be negative if a large amount of redundant information swamps the synergistic part [15]. Recently, Mediano, Rosas, et al. introduced a complete decomposition of the excess entropy for a multivariate system. Based on the partial information decomposition framework [16,17], the *integrated information decomposition* (ΦID) [13,15] reveals how all of the elements of **X** (and ensembles of elements) collectively disclose temporal information about each-other. A rigorous derivation of the framework is beyond the scope of this paper, but intuitively the ΦID works by decomposing E(X) into an additive set of non-overlapping values called “integrated information atoms”, each of which describes a particular dependency between elements (or ensembles of elements) at time t−1 and time *t*. For example, the atom $\left\{X^{1}\right\}\to \left\{X^{1}\right\}$ refers to the information that $X^{1}$ and only $X^{1}$ communicates to itself and only itself through time (“information storage”). Similarly, $\left\{X^{1}\right\}\to \left\{X^{2}\right\}$ is the information that $X^{1}$ uniquely transfers to $X^{2}$ alone. More exotic combinations also exist, for example $\left\{X^{1}\right\}\left\{X^{2}\right\}\to \left\{X^{2}\right\}$ which is the information redundantly present in $X^{1}$ and $X^{2}$ simultaneously that is “pruned” from $X^{1}$ and left in $X^{2}$.

The ΦID framework allows us to refine the original heuristic Φ(X) and construct a more rigorous metric for emergence: for a two-element system, the atom $\left\{X^{1},X^{2}\right\}\to \left\{X^{1},X^{2}\right\}$ is the synergistic information that the whole (and only the whole) discloses about its own future (Mediano et al. refer to this as “causal decoupling”, although we will refer to it simply as temporal synergy). It can be thought of as measuring the persistence of the whole qua itself without making reference to any simpler combinations of constituent elements. The ΦID framework also provides another emergence-related metric “for free:” Mediano et al. claim that “downward causation” [18,19] is revealed by atoms such as $\left\{X^{1},X^{2}\right\}\to \left\{X^{1}\right\}$, where the synergistic past of the whole informs on the future of one of its individual constituent atoms. In this paper we focus on the temporal synergy, but all of the same technical concerns (localization, incongruous dynamics, etc) also applies to the downward causation atoms.

For a two-variable system evolving through time (such as our example macro-scale $\tilde{X}$), it turns out that there are only sixteen unique integrated information atoms, and they are conveniently structured into an elegant lattice (see Figure 2). This means that given some “double-redundancy” function that can solve for the bottom of the lattice ($\left\{X^{1}\right\}\left\{X^{2}\right\}\to \left\{X^{1}\right\}\left\{X^{2}\right\}$), it is possible to bootstrap all the other atoms via Mobius inversion. For the given lattice A, the value of every atom ($\Phi_{\partial }\left(A\right)$) can be calculated recursively by:(15)$$\Phi_{\partial }\left(A\to B\right)=I_{\tau sx}\left(A\to B\right)-\sum_{\begin{array}{c} A^{\prime }\to B^{\prime }\\ ⪯\\ A\to B \end{array}}\Phi_{\partial }\left(A^{\prime }\to B^{\prime }\right)$$
where $I_{\tau sx}$ is the double redundancy function proposed by Varley in [11], which quantifies the shared information across time (for a more detailed discussion, see the above citation). The Mobius inversion framework provides an intuitive understanding of the double synergy atom $\left\{X^{1},X^{2}\right\}\to \left\{X^{1},X^{2}\right\}$: it is that portion of E(X) that is not disclosed by any simpler combination of elements in X then X itself. The system $\tilde{X}$ shown in Figure 3 (upper right) shows non-trivial integrated emergence, with a value of $\Phi_{\partial }\left(\left\{X^{1},X^{2}\right\}\to \left\{X^{1},X^{2}\right\}\right)\approx 0.031$ bit.

## 3. Incongruous Emergence Appears in Multiple Frameworks

Having described how both formal approaches to emergence choose to define it, we are equipped to discuss local *incongruous* emergence. As mentioned above, incongruous dynamics occurs when, locally, the information structure of a system is the opposite of the expected tendency. Even if the system on average displays some emergent dynamic, it does not necessarily follow that it displays emergent properties at every moment. Below, we will show how the localizability of excess entropy implies incongruous dynamics can occur in both coarse-graining and synergy-based emergence frameworks.

### 3.1. Flickering Emergence in Coarse-Graining Approaches to Emergence

We say that a system **X** admits an emergent macro-scale $\tilde{X}$ if $\log(\bar{F}\left(\tilde{X}\right)/\bar{F}\left(X\right))>0$, where $\bar{F}\left(X\right)$ is the effectiveness of **X** (Equation (Equation 11)). In the context of the local excess entropy, the normalization by log(N) does not necessarily make sense (the inequality that E(X)≤log(N) only holds in the average case: the local mutual information is unbounded), however, we can instead consider the signs of the relevant local excess entropies. When a change in sign occurs, information that may have been informative at one scale (i.e., helps us make better predictions about the future) may be misinformative at another (i.e., pushes us towards the wrong prediction).

We say that incongruous dynamics occurs when $e(\tilde{x})<0$ and e(x)>0. That is, when a transition that is informative at the micro-scale gets mapped to a transition that is misinformative at the macro-scale. From the perspective of a scientist attempting to understand a complex system, this would occur if, on average more predictive power (or controllability) is accessible at the macro-scale, but there exist a subset of transitions at the micro-scale that actually do worse when coarse-grained. When considering the system in Figure 1 and its associated macro-scale, we can construct the local excess entropy for every transition (visualized in Figure 3).

It is visually apparent that the large number of weakly informative transitions at the micro-scale are getting mapped to weakly misinformative transitions at the macro-scale. In fact, ≈52.73% of informative micro-scale edges map to misinformative macro-scale edges! This means that, even though our toy system **X** displays non-zero emergence on average, over half of all informative micro-scale transitions are mapped to misinformative macro-scale transitions during coarse-graining. These are generally lower-probability transitions however, which accounts for the overall display of emergence. By running a random walk on X and computing the ratio $e\left({\tilde{x}}_{t}\right)/e\left(x_{t}\right)$ we can see a number of instances where the sign is negative because incongruous dynamics have occurred (indicated with grey arrows). This is an example of what we call “flickering emergence”, where the emergent quality transiently falls apart, like a candle sputtering.

#### Application to Networks

The phenomenon of incongruous dynamics is also relevant to extensions to other domains, such as network science. Klein, Griebenow, and Hoel showed that applying the same framework to the dynamics of random walkers on complex networks could reveal informative higher scales in the network structure, which in turn could be linked to many aspects of graph theory and network science [20,21]. In this approach, nodes in a network are aggregated into “macro-nodes” (analogous to communities), and those macro-nodes collapsed into a simpler network. From there, the effectiveness of the micro-scale network can be compared to the effectiveness of the macro-scale network in the usual way. This topological emergence has been found in a variety of biological networks and associated with evolutionary drives towards robust, fault-tolerant structures [21].

In keeping with prior work, Klein et al. focused on the average information structure over the entire network, however, we can do the same kind of localized analysis on the network that we do on the Markov chains. In Figure 4 (left panel), we can see a structural connectivity matrix taken from a random subject in the Human Connectome Project data set [22] (previously studied in [23]) from which we have computed the local excess entropy associated with each edge. We then used the Infomap algorithm [24,25] on the raw structural connectivity network and coarse-grained the communities into macro-nodes. When considering the macro-scale transition probability matrix, we can see that the main diagonal (corresponding to a walker staying put) are overwhelmingly informative, while off-diagonal transitions are generally misinformative (Figure 4, centre panel). This is consistent with the intuition that, in state-transition networks, Infomap finds shallow, metastable transient attractors in the landscape [26].

While the effectiveness of the macro-scale was barely lager than the effectiveness of the micro-scale (≈0.37 bit at the micro-scale vs. ≈0.38 bit at the macro-scale), we find that incongruous dynamics is quite common in this particular network: ≈22.47% of informative micro-scale edges are mapped to macro-scale edges that are misinformative. To assess whether the distribution of informative and misinformative micro-scale edges related to the overall macro-scale structure of the network, we examined the distribution of informative and misinformative edges within and between communities/macro-nodes. We found that informative edges were roughly just as likely to link within-community nodes (≈52.3%) as between community nodes (≈47.7%), however, misinformative edges where overwhelmingly more likely to link disparate communities (≈80.4%) as opposed to fall within a single community (≈19.6%). To ensure that these results held generally and were not an artifact of the Infomap algorithm, we replicated these results using a spinglass [27,28] and a leading eigenvector community detection algorithm and found that the results were consistent (see Appendix A). All analyses were done using the Python-iGraph package [29]. These results collectively show that, in much the same way expected effective information and emergence relate in fundamental ways to network topology, so do their local counterparts.

### 3.2. Flickering Emergence in ΦID-Based Approaches to Emergence

In the ΦID-based framework, emergence is associated with information about the future of the “whole” that can only be learned by observing the whole itself and none of its simpler constituent components. This value can be computed using an decomposition of the excess entropy [11,13]. Like the coarse-graining framework, the decomposition of the excess entropy can be localized to particular moments in time. For a given transitions ($\left(x_{t-1}^{1},x_{t-1}^{2}\right)\to \left(x_{t}^{1},x_{t}^{2}\right)$) with local excess entropy e(x), we can construct a local redundancy lattice a and solve it with the same Mobius inversion. All that is required is that the redundancy function is localizable in time. The function used above, $I_{\tau sx}$ is localizable and can be used to construct a local integrated information lattice for any configuration of X [11].

Once again, we take advantage of the distinction between informative and misinformative local mutual information for our indicator of incongruity. Here we say that incongruous dynamics occurs when the expected synergistic flow ($\left\{X^{1},X^{2}\right\}\to \left\{X^{1},X^{2}\right\}$) is informative (positive), but the local value of the same atom is negative. Intuitively, this is the case when, on average, the future of the whole can be meaningful predicted from its own past; however, there are some configurations where the current global state would lean an observer to make the wrong prediction about the future global state.

When considering the example macro-scale system (Figure 3), we found that 31.25% of the sixteen possible transitions had a negative value for the double-synergy atom. In Figure 5, we can see the global integrated information lattice for two different transitions, both landing in the universal-off state ((0,0)). It is clear that these two transitions have radically different local information structures, including opposite-signed double-synergy, double redundancy, and “downward causation”, atoms: the expected value of any given atom elides a considerable amount of variability in the instantaneous dependencies. We can see this manifesting as flickering emergence when we run a random walk on the system and perform the ΦID for every time-step (Figure 5, Bottom): the system transiently moves through periods of informative, and misinformative emergence depending on the specific transitions occurring.

## 4. Discussion

In this work, we have introduced the notion of “flickering emergence”, which describes how a system can, on average, admit a meaningful emergent dynamic that falls apart locally when considering particular transitions. We argue that this is likely a feature of any formal approach that is built on the Shannon mutual information between the past and the future (excess entropy [12]). To demonstrate this phenomenon, we assess two approaches which share the same core feature of excess entropy, but define “emergence” in very different ways. The first takes a coarse-graining approach, comparing micro- and macro-scales [1,6,7,20], while the second takes a synergy-based approach and looks for information in the “whole” that is not accessible when considering any of the “parts [2,8].

The purpose of this paper is not to “poke holes” in, or critique, either approach to emergence. Instead, our aim has been to highlight how the particular mathematical formalism one commits to can produce intriguing new properties that may not have been obvious from the outset. In this case, the commitment to a temporal mutual information-based approach necessarily raises the question how to think about local instances and what it might mean for a measure of emergence to be locally negative.

How we interpret incongruous dynamics and flickering emergence depends largely on how we interpret “emergence” as a concept. If emergence refers solely to an observer’s ability to model some process (as in [30]), then incongruous dynamics may just reflect the limitations inherent in modelling: no model makes perfect predictions in all contexts. For example, one can use a coarse-grained model of the heart as an oscillating pump very successfully and generally ignore the particular myocardial cells *when the heart is healthy*, however pathological states, such as ventricular fibrillation, can cause the predictions of the heart-as-a-pump model to diverge dangerously from the micro-scale model. In this case, the predictions of the macro-scale model fail and the apparent “emergent” structure of the pump falls apart.

If, instead of being a reflection of a modellers perspective, “emergence” is treated as an ontologically “real” process, and mapped to other observable phenomena, the situation may get more complicated. Consider the recent proposal from Luppi et al. [31], who suggest a link between integrated emergence and the persistent sense of one’s conscious self as an unitary agent (also discussed in [32]). If there is an identity between temporal mutual information and the sense of self (an observable phenomena), then we are forced to wrangle with the question of temporal locality. The flickering of emergent synergy would imply some kind of associated flickering of the sense of self, or conscious awareness, which should be an observable (or at least reportable) process amenable to study.

Part of the difficulty with interpreting flickering emergence comes from the fact that it does not necessarily define *why* a particular moment breaks from the long-term behaviour. In some cases, it may merely be statistical fluctuations about the mean, while in others it might be reflective of an underlying change in the causal structure of the system. Any interpretation must be done in the context of the particular system under study and leverage domain expertise beyond what is accessible just from the statistics alone.

In addition to the main question about local information dynamics, a second aim of this paper was to bring the two frameworks into more explicit dialogue. Both approaches have developed largely in parallel (despite having a common intellectual heritage in integrated information theory) and propose different notions of what it means to be emergent. As we have seen however, they share significant commonalities in spite of their differences. While “emergence” is typically discussed as a single phenomena in complex systems, there is a strong argument to be made for the “pragmatic pluralist” approach [33,34], in which many different “kinds” of emergence are recognized in parallel. Under such a framework, both frameworks discussed here would both be considered valid kinds of emergent dynamics: similar enough to belong to a common category of phenomena, but distinct enough to be considered separate. This is very similar to how the question of defining “complexity” has practically been resolved: historically, there has been considerable debate on how we might define “complexity” in “complex systems”, and various measures have been proposed over the years, including algorithmic compressibility [35], entropy rate [36], and integration/segregation balance [37]. Despite the considerable ink spilled on the question, Feldman and Crutchfield argued that there likely is not a single measure of what it means to be “complex” and that mathematical attempts at a universally intuitive measure were misguided [38]. Instead, the field has largely moved towards an understanding that different notions of “complexity” are appropriate in different contexts and can illuminate different aspects of system dynamics, all of which may be considered “complex” in their own way (for example, see [26], which proposes the notion of a “dynamical morphospace” to characterize systems along different axes complex dynamics). A similar resolution may end the perennial conflict between what is and is not a valid measure or kind of emergence. The multi-scale framework may turn out to be useful when attempting to think about how cells can be coarse grained into tissues (where there is a natural distinction of scales), while the integrated information framework may turn to be useful when considering the computational properties of ensembles of neurons [11] or flocking objects [8], but not vice-versa. Both of these may be instances where reductionism fails to provide the crucial insight into a collective process, but importantly, reductionism may fail *for different reasons in different contexts*.

Finally, we should note that there are other mathematical frameworks for tackling emergence in dynamical systems that are not based on classical information theory, and do not necessarily assume a fundamentally stochastic dynamical system (see [39] and associated papers). While stochastic models are ubiquitous in science due to the uncertainty inherent in modelling data, if the generative processes are actually deterministic, probabilistic, entropy-based models may be misleading about the causal structure of the system under study [40]. In such cases, measures of emergence based on algorithmic complexity may be more appropriate, such as the proposed measure given in [41]. There, Abrahao et al. describe an algorithmic approach to understanding emergence in finite, deterministic, dynamical systems based on trajectories through a state-space and the effects of perturbation on individual updates. The possibility of update-specific analysis implies that time-resolved analysis of emergent dynamics may not be restricted to frameworks based on Shannon information theory and expected values. Future work comparing and contrasting frameworks based on Shannon information theory and frameworks based on algorithmic information theory will help further our understanding of causal emergence in complex systems.

In summary, we feel that the problem of information-based approaches to emergence remains a rich area of research, with considerable territory remaining to be explored. Unexpected phenomena, such as flickering emergence may force us to challenge how we think about emergent properties in complex systems and potentially inform future research directions. The richness and variability of different measures is, to our mind, a feature, rather than a bug, with intriguing commonalities and differences.

## 5. Conclusions

Regardless of whether one prefers coarse graining-based or synergy-based approaches to emergence, and whether one chooses to think of emergence purely as a question of modelling, or of truly novel physical properties, any approach based on excess entropy is likely to display both incongruous dynamics and flickering emergence. The localizability of mutual information and related measures (such as the transfer entropy) shows us that, even when emergence (however defined) occurs on average, there can still be complex and unexpected moment-to-moment deviations from that average. Some of those deviations can run in express opposition to the expected behaviour (signified by negative local mutual information). These deviations from the long-term norm may have profound implications for how we think about emergent properties in nature, and suggest new avenues of research for scientists interested in the role emergence plays in the natural world.

## Figures and Tables

**Figure 1 entropy-25-00054-f001:**
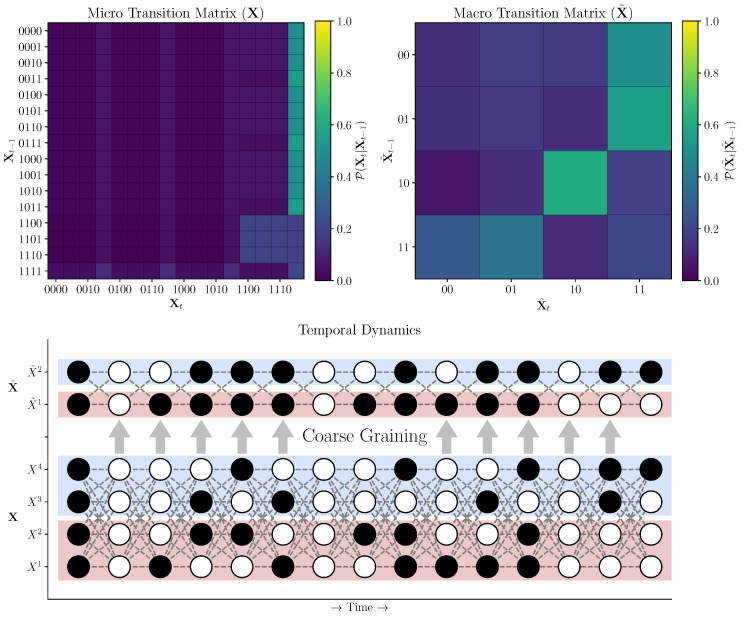
**Transition probability matrices for a micro- and macro-scale system. Top Left:** The transition probability matrix for the micro-scale view of system **X**. Each row corresponds to a possible past state $X_{t-1}$, and each columns corresponds to a possible future state $X_{t}$. The value of each cell TPM${}_{ij}$ gives the conditional probability of transitioning to future state $x_{t}$ given that the system is in state $x_{t-1}$. **Top Right:** The transition probability matrix for the macro-scale of **X** (denoted as $\tilde{X}$). **Bottom:** A cartoon showing the coarse-graining procedure. The micro-scale boolean network model **X** has four, binary, interconnected elements ($X^{1-4}$), which update their states according to the micro-scale TPM as the system evolves through time (each “column” corresponds to the state of every element in **X** at a given moment in time). When coarse graining, pairs of nodes are grouped together using a logical-AND gate. The two micro-scale nodes $X^{1}$ and $X^{2}$ get mapped to macro-scale node ${\tilde{X}}^{2}$, as indicated by the blue colour, and likewise for the red-labelled nodes. The state-transition dynamics of the macro-scale system is governed by the macro-scale TPM.

**Figure 2 entropy-25-00054-f002:**
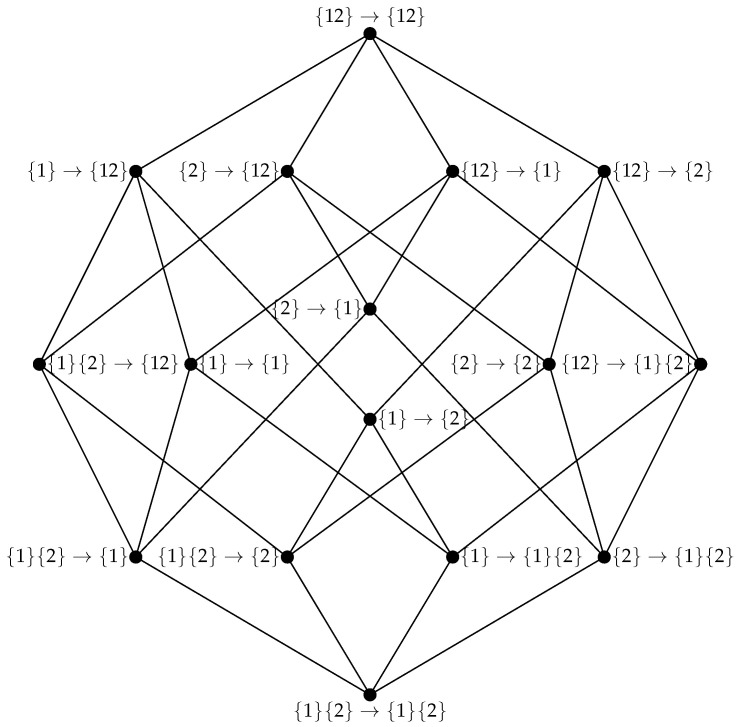
The double-redundancy lattice for a system $X=\{X^{1},X^{2}\}$. Every vertex of the lattice corresponds to a specific dependency between information in one element (or ensemble of elements) t−1 and another at time *t*. We use index-only notation following Williams and Beer [13,16] for notational compactness.

**Figure 3 entropy-25-00054-f003:**
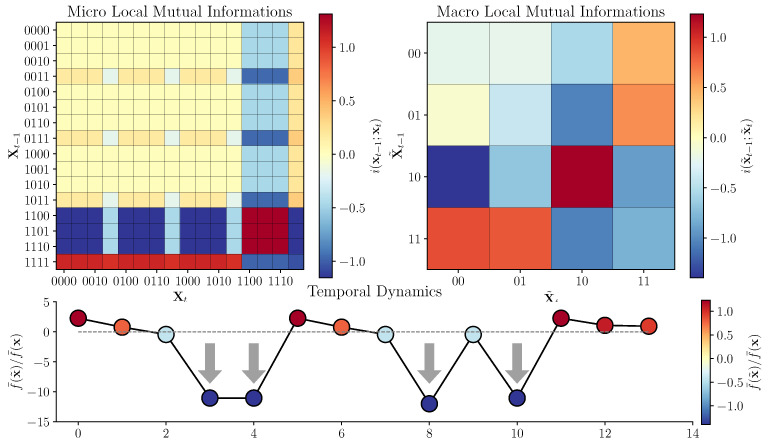
**Local information of transitions at macro- and micro-scales. Top Left:** The local mutual informations associated with every transition at the macro scale. We can see a mixture of informative and misinformative transitions distributed over the matrix. **Top Right:** The local mutual informations associated with every transition at the micro-scale. We can see that a very large number of weakly informative transitions at the micro-scale (upper left-hand square) get mapped to misinformative transitions at the macro-scale (upper left-hand square of the macro-scale matrix). **Bottom:** A time series generated by a weighted random walk on the micro-scale transition matrix. We plot the ratio of the local mutual information of each transition at the micro-scale to the associated transition at the macro-scale. If the microscale is informative and the macroscale is misinformative, we say that that transition displays “incongruous dynamics”. This “incongruous dynamics” shows that, while on average the macro-scale may be more informative than the micro-scale, there are a large number of micro-scale transitions that become not only less informative, but actively misinformative!

**Figure 4 entropy-25-00054-f004:**
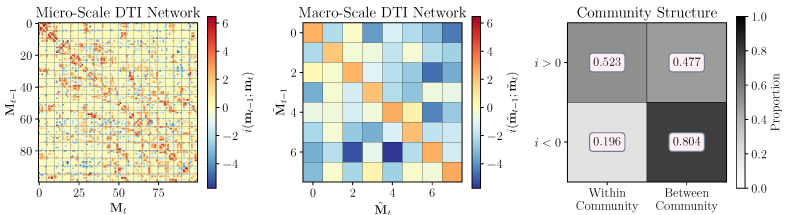
**Incongruous dynamics in complex networks. Left:** A single structural connectivity matrix taken from the Human Connectome Project [22], coloured according to the local excess entropy associated with a random walker on each edge. **Middle:** The same transition probability matrix after renormalizing the micro-scale community structure with Infomap [24,25]. **Right:** The proportion of informative and misinformative micro-scale nodes that sit within, or between, a macro-scale community.

**Figure 5 entropy-25-00054-f005:**
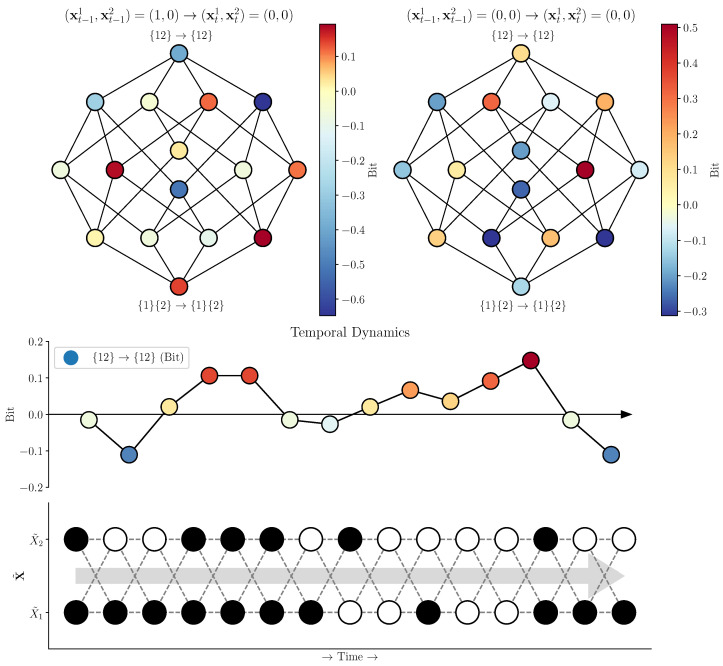
**The structure of temporal mutual information varies across time. Top:** The two double-redundancy lattices show the local integrated information decompositions for two distinct transitions ((0,1)→(0,0) and (0,0)→(0,0)). Despite having the same end-state, the information structure of these two transitions is completely different. One transition has an informative causal decoupling, while the other has a misinformative one. As well as opposite-signed double-redundancy ({1}{2}→{1}{2}) and a number of other discrepancies. **Bottom:** A visualization of the phenomenon of “flickering emergence” in the context of ΦID. As the system $\tilde{X}$ evolves over time, it cycles through states (lower plot), and for each of those transitions, we can calculate the instantaneous causal decoupling (upper plot). We can see that incongruous dynamics can occur at different times, interspersed between congruent emergence.

## Data Availability

No data was collected for this study. Python scripts are available from the author upon request.

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
