# Peer review of "Flickering Emergences: The Question of Locality in Information-Theoretic Approaches to Emergence"

_entropy, 2022, doi:10.3390/e25010054_

Round 1

Reviewer 1 Report

See attached pdf.

Author Response

z

Reviewer 2 Report

The paper is well written and well organised but it is missing some key developments to be inserted in the current area discourse.  In particular the recent work of Abrahao:

https://royalsocietypublishing.org/doi/10.1098/rsta.2020.0429

and Hernandez:

https://direct.mit.edu/artl/article-abstract/24/1/56/2880/Undecidability-and-Irreducibility-Conditions-for?redirectedFrom=fulltext

And the work of the Tegner group:

https://www.cell.com/iscience/fulltext/S2589-0042(19)30270-6#%20

In the literature above missing in the overview, the authors use a refinement/generalisation of information theory as they have proven that Entropy-only approaches can be deceiving:

https://journals.aps.org/pre/abstract/10.1103/PhysRevE.96.012308

I will support the publication of this work therefore if the above literature is discussed and the methods of the authors are put in the context of this work, also in the context of the perturbation analysis over time of the paper above published in iScience with some similar network approaches to that of this paper authors.

Author Response

See PDF

Round 2

Reviewer 1 Report

The author seems to truly have taken my recommendations seriously, and I’m thrilled to say that this work is fit for publication. I commend the author for their patience and careful responses to my comments, and I want to congratulate the author on this interesting development.

P.S. The figures are phenomenal, and I can see them being used frequently in lectures and pedagogical materials in the future. Cheers.

Reviewer 2 Report

I am satisfied with the changes and I support its publication.